**DOI: 10.1038/ncomms12971**　　**OPEN**

# Forward design of a complex enzyme cascade reaction

Christoph Hold[1], Sonja Billerbeck[1,†] & Sven Panke[1]

Enzymatic reaction networks are unique in that one can operate a large number of reactions under the same set of conditions concomitantly in one pot, but the nonlinear kinetics of the enzymes and the resulting system complexity have so far defeated rational design processes for the construction of such complex cascade reactions. Here we demonstrate the forward design of an *in vitro* 10-membered system using enzymes from highly regulated biological processes such as glycolysis. For this, we adapt the characterization of the biochemical system to the needs of classical engineering systems theory: we combine online mass spectrometry and continuous system operation to apply standard system theory input functions and to use the detailed dynamic system responses to parameterize a model of sufficient quality for forward design. This allows the facile optimization of a 10-enzyme cascade reaction for fine chemical production purposes.

[1] Department of Biosystems Science and Engineering, ETH Zürich, Mattenstrasse 26, 4058 Basel, Switzerland. † Present address: Department of Systems Biology, Columbia University, 1130 St Nicholas Avenue, New York, New York 10032, USA. Correspondence and requests for materials should be addressed to S.P. (email: sven.panke@bsse.ethz.ch).

The ability to simultaneously operate reaction networks in one pot is highly attractive as large modifications of molecular structure or perfect optical purity can be achieved in one processing step, including reactions that in isolation would be thermodynamically unfavourable. With increasing network complexity, this ability becomes more and more exclusive to the biochemical reaction domain, as only nature has provided a huge set of catalysts that operate under similar conditions. Correspondingly, many such networks are operated in cells[1–4]. However, while acquiring enzymes remains laborious (but can be much facilitated by exploiting thermostable enzymes[5–8]), cell-free reaction networks or 'cascade reactions' offer the advantage of the absence of membrane-induced mass transfer and many toxicity effects, facilitated use of non-natural compounds, use of not (exclusively) aqueous solvents, increased flexibility in network structure[6,9] and better control of the reaction[10–14]. Consequently, such cell-free reaction networks or cascade reactions have been broadly distributed for a variety of purposes, including the synthesis of (activated) mono- and oligosaccharides[15–19], various fine chemicals[14,20–26], monomers[8,27,28], polymers[29–32], fuels[6,7,33], hydrogen[34,35] and the generation of electricity[36,37].

However, increasing complexity often leads to non-optimal behaviour as the interactions remain poorly understood, and therefore the systems remain difficult to scale. First attempts at a semi-rational system optimization have been undertaken[21,38], but a design process is best supported by a full system model that is well enough parameterized to reflect the main aspects of the behaviour of the cascade reaction. However, enzymes are subject to nonlinear kinetics, such as Michaelis–Menten-type kinetics, which is often complicated by feedback, cooperative or allosteric elements. This makes the development and in particular the robust parameterization of a suitable model a challenge, as standard experiments are not sufficient to resolve the many instances of non-identifiability that can accompany the parameterization efforts. Therefore, the forward design and implementation of synthetic biochemical pathways has not been demonstrated yet.

Interestingly, because of the importance of kinetics for understanding intracellular metabolism, most efforts towards the establishment of models for large enzyme reaction networks were undertaken for *in vivo* systems[39,40], which represents an even more challenging system because of the additional mass-transfer barriers ((intra)cellular membranes) and the variable composition of the reaction system (cellular response to environmental stimuli). Consequently, though dynamic models exist for sections of central carbon metabolism *in vivo*, they are of limited use for forward engineering since during their development it was not possible to perform sufficiently dynamic experiments, such as applying diverse (intracellular) perturbations with different compounds and measuring a sufficient number of compounds. But even in cell-free systems, such as *in vitro* oscillators[41–44], but also cascade reactions[45], model development does not go beyond the estimation of a few parameters, and thus leaves the complexity of the system unresolved and thus design uncertain.

We reasoned that *in vitro* forward design would become possible if the experimental system allowed a sufficiently broad application of dynamic challenges and a sufficiently detailed recording of the system's responses. We therefore explored the construction of a highly versatile experimental set-up that allows the generation of standard input functions from systems theory applied to a continuous stirred tank reactor (CSTR) and the collection of sufficiently detailed concentration time series from the system's response with a recently developed real-time mass spectrometry method[38]. While a CSTR would not be a suitable reactor for large-scale implementation of a multi-step reaction, we reasoned that the set-up would in fact remove the central obstacle for forward engineering of complex reaction systems, namely limited model scope due to insufficient experimental data quantity and quality, and thus enable generating a model for design. We demonstrate this by successfully forward engineering a 10-enzyme cascade reaction to produce an important intermediate in enzyme-catalysed monosaccharide synthesis, dihydroxyacetone phosphate (DHAP)[46].

## Results

**Tracking complex system perturbations with high data density.** Implementation and analyses of the reaction system were performed in a CSTR (Fig. 1a)[47], which is indispensable for a thorough engineering analysis of the system. The composition of the feed into the reactor and the dynamics of its change were set by controlling multiple feed high-performance liquid chromatography (HPLC) pumps and an injector loop. This allows freely impressing diverse concentration feed profiles, representing different standard input functions, onto the reaction system (Fig. 1b). To record the dynamics of the response of the reaction system to the input functions, the constant product stream is removed through an ultrafiltration membrane, which retains the enzymes (thus stopping the reactions) but lets pass the liquid with remaining starting materials, intermediates and products. This continuous reactor effluent is conditioned with MS matrix buffer for subsequent online measurement in an electrospray ionization (ESI) MS (Supplementary Table 2), which is operated in multi-reaction monitoring mode and allows the determination of the concentration of one compound every 500 ms (ref. 38). Consequently, even reaction systems can be easily tracked, allowing systems in the order of 20 compounds to be analysed in ~20 s (including standards). To separate the dynamics of the system from contributions of the set-up, we thoroughly identified the transfer function of the experimental set-up in step-wise fashion (Fig. 1c and Supplementary Fig. 1) and found that the influence of the set-up can be accurately described as a coupled system of three CSTR's describing pump, reactor and post-reactor dilution elements.

Next to this identification of the set-up, we ensure accurate measurement of compound concentrations despite the potential for ion suppression due to the concomitant entry of many compounds into the ESI chamber of the MS. We use chemically orthogonal standards to measure the ratio of the flows from effluent and conditioning and then either isotopologues or an orthogonal standard with extensive calibrations to calculate compound concentrations (Supplementary Fig. 2). This set-up can be used for a broad variety of cascade reactions involving at least 40 compounds, or multiples of this number if the effluent is split and directed to different mass spectrometers. We also ensured sufficient enzyme stability for our specific reaction system (Supplementary Fig. 3).

**Enabling structural model identification.** As a first test with still limited scope we tested the set-up's capacity for structural model identification. Interestingly, the reaction mechanism of a variety of glucokinases (Glk's, for abbreviations of enzyme and compound names see Supplementary Table 1), such as that of the yeast *Saccharomyces cerevisiae*[48], is described to include an inhibition term for ADP, while the mechanism for the enzyme of *Escherichia coli* that was used in the present experiments (Supplementary Table 3), does not (Supplementary Note 1 and Supplementary Table 24). Therefore, we impressed different input functions representing very different substrate concentration dynamics onto the reactor containing only Glk and

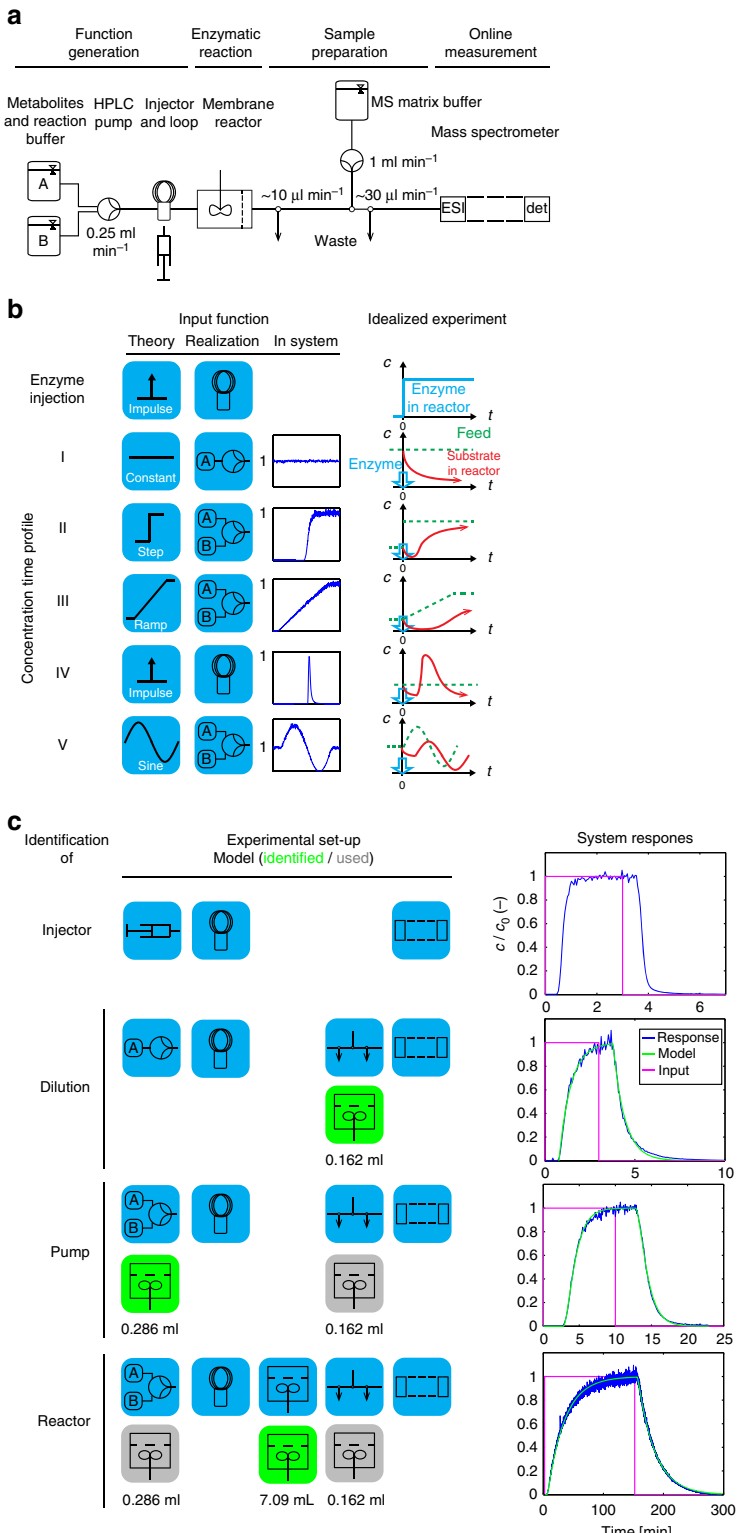

**Figure 1 | Experimental strategy for system identification.** (**a**) Set-up of function generation unit consisting of one or multiple HPLC pumps, injection loop and/or syringe port, enzymatic reaction consisting of a CSTR with membrane for enzyme retention, sample preparation to condition continuous reactor effluent for MS injection and online measurement in a ESI-triple-quad MS. (**b**) Possible input functions with theoretical form, practical realization of the function and measured system response at the outlet of the CSTR, and idealized substrate profile in CSTR without (green stippled line) and with (red solid line) enzyme reaction system. (**c**) Stepwise full identification of the dynamics of the set-up. Model: schematic representation of the stepwise identification process. Each line represents a schematic of the specific set-up used for identification of the specific element (known elements in grey, element to be identified in green). Once identified, a system element was modelled as indicated in the subsequent identification steps. Experiment: result of the particular identification with the representation of the generated perturbation in magenta and the system response in normalized concentration as simulated (green, using the values indicated in 'Model') and measured (blue).

used the recorded dynamic responses to estimate the parameters when the inhibition term was not included (Fig. 2a and Supplementary Fig. 4). Clearly, despite the limited number of parameters the selected optimizer cannot identify a suitably small range of parameter values for the affinities for Glc and ATP, even when the impressed substrate profiles become complex. This suggests non-identifiability due to the assumption of a wrong model. Including the ADP-inhibition term allows estimating a suitable parameter set (Fig. 2a, bottom panel), but only after the input function has become complex enough to resolve structural non-identifiability due to too simple feed profiles.

**Assembly of a model cascade reaction**. Next, we implemented a multi-step reaction system from purified enzymes (mostly commercial and from different hosts, Supplementary Table 3) to produce DHAP from glucose (GLC) as shown in Fig. 2b. The system contains 10 enzymes and 17 compounds, and is thus in terms of size at the upper end of cascade reactions that are intended for cell-free chemical production[6,35]. We chose this pathway because, first, it generates an essential precursor for enantioflexible synthetic routes (DHAP can be converted into a stereochemically complete set of vicinal diols[46]); second, DHAP synthesis and cofactor regeneration can be achieved by building essentially on glycolytic enzymes, whose mechanisms are sufficiently well known to develop a mechanistic dynamic model, but their interactions are complex because of multiple regulatory feedback loops; and third the need for ATP and NAD recycling in the pathway reflects that the thermodynamic profiles of cascade reactions are rarely monotone and activation by cofactors is often necessary. We also expanded this network by an additional reaction, reduction of DHAP by glycerol 3-phosphate dehydrogenase (G3d) to glycerol 3-phosphate (G3P) as a model product. This reflects that DHAP is rather an essential precursor than a final product[18]. As the G3d reaction regenerates NAD, lactate dehydrogenase (Ldh) can be omitted in those cases.

**Model formulation and parameterization**. We translated this reaction system into a model. By balancing the 18 compounds in the previously identified 3-vessel system (Fig. 1c), an overall model with 54 states describing the generation of input profiles and concentration propagation through the system was derived. We also derived 11 mechanistic enzyme rate laws for the enzymes of the reaction system based on literature-described reaction models (Supplementary Note 1 and Supplementary Table 24) with 60 parameters, such as affinities and Hill coefficients. The values of these parameters were not known *a priori* and depend on the specific enzyme and reaction conditions such as pH, buffer composition, temperature or ionic strength; even reported values for some of the enzyme parameters vary by orders of magnitude. We therefore estimated these parameters by dividing the system into manageable subsystems of up to 4 enzymes and maximally 24 parameters (Supplementary Fig. 5).

For each subsystem, we conducted different types of perturbation experiments ranging from pulsing enzymes via a constant feed to substrate gradients, resulting in a total of 22 experiments of the type shown in Fig. 2c (Supplementary Figs 6–11 and Supplementary Tables 4–17). While simulations of concentration time series of separate subsystems using parameters that were derived from the experiments specific for this subsystem are generally in excellent agreement with the data (Fig. 2c), the quality of the simulations of groups of subsystems with such parameters is often less satisfactory. Therefore, some of the subsystems required re-estimation of some enzyme parameters multiple times. Ultimately, the set of 22 experiments allowed estimating a final set of parameters (Supplementary Table 18 and

Fig. 2d) that reproduced all experiments well as the basis for the subsequent design phase. Only few parameters (for example, for Pfk) are at the boundary of the allowed range, suggesting cases of limited identifiability that might have been resolved with additional experiments. Next, consultation of suitable databases (in particular BRENDA[49]) allowed comparing 32 of the 36 compound affinities, whose knowledge we had deemed most uncertain before, with previously known values. Of these 32 affinities, 24 are within generally reported ranges (Fig. 2d). Possible reasons include that our reaction conditions might have been different from those under which these parameters had been previously established (in particular, presence of a broad variety of compounds). However, we refrained from further refining the perturbations and the parameter set as the models for the subsystems, as well as the complete model are already fully capable of enabling forward design.

**Optimizing a cascade reaction for product concentration**. To demonstrate this, we optimized different aspects of the reaction cascade, specifically enzyme use in the upper and in the lower part of the cascade separately, as well as in the cascade as a whole, and, finally, the use of the most expensive cofactor in the system, NAD. For the optimization of enzyme use, this meant identifying the optimal distribution of a given total amount of enzyme over the different reaction steps. Please note that, fundamentally, such a cascade would ideally be run in a batch or fed-batch reactor to prevent loss of intermediates (and thus a reduction in yield on GLC). However, to maintain the ability to track the behaviour of the optimized system accurately and at high time resolution, we remained in the CSTR setting, which is anyway close to a batch scenario as the dilution rate is rather low ($2.1\,h^{-1}$). We started with the upper part of the cascade (from GLC to fructosebisphosphate (FBP)) and compared the scenario in which each of the three enzymes was available with the same activity ('equi-activity scenario', total enzyme amount 3 U) to the model-predicted optimal distribution (Fig. 3a and Supplementary Table 19). Clearly, prediction and experiment for the optimized scenario are in good agreement, and FBP concentration in the effluent is increased by 26%, even though the GLC consumption remains nearly constant. Next, we optimized the lower part of the cascade (starting with FBP, total of 7 U, Supplementary Table 20) for G3P and pyruvate (PYR) production, again in comparison with the equi-activity scenario. Again, prediction and experiment show good agreement in dynamics and steady-state concentrations and a large increase in G3P concentration in the effluent by 75% (Fig. 3b). Finally, we optimized the entire cascade (GLC to G3P and PYR) with different total amounts of enzymes to be optimally distributed (20 and 40 U). This allows, in good agreement with the predictions, a steady increase in GLC consumption and G3P and PYR production (for example, 88% increase of G3P steady-state concentration in the 20 U scenario) (Fig. 3c and Supplementary Table 21).

**Optimizing a cascade reaction for cofactor concentration**. Next to productivity, cofactor costs are a major impediment to the implementations of cascade reactions. Consequently, we analysed how far the NAD concentration could be reduced without reducing the G3P productivity shown in Fig. 3c. We conducted *in silico* experiments with the NAD concentration changing in steps from 0.1 to 2 mM (Supplementary Fig. 12), and found that an NAD concentration of 0.25 mM, only one-fourth of the previously used concentration, is predicted to be sufficient for comparable product formation (Supplementary Table 22). When the corresponding experiment at reduced NAD concentration is

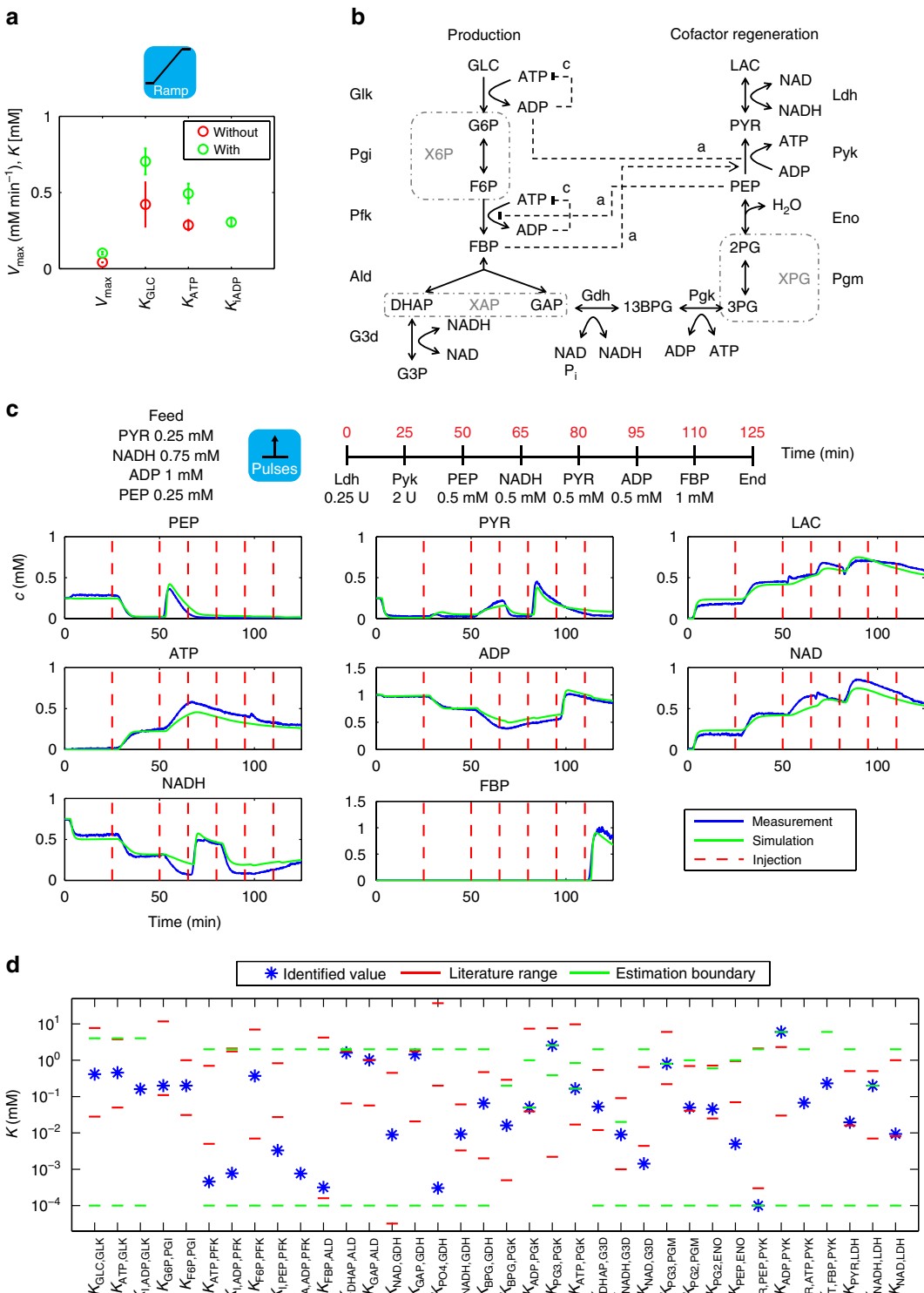

**Figure 2 | Model identification.** (**a**) Structural identification of the reaction mechanism of Glk. Displayed is on the top a schematic of the type of experiment used for identification. The data were used to estimate the parameters of two different rate equations for Glk, once excluding (red symbols) and once including (green symbols) a term for ADP inhibition. We carried out for each model 100 independent parameter estimation runs and show the obtained s.d.'s for the parameter estimation, which suggest the requirement for an ADP inhibition term. (**b**) Enzymatic cascade reaction for the production of DHAP. Note the simulated consumption reaction for DHAP by G3d-catalysed conversion to G3P. Stippled arrows: enzyme activation. Blunt stippled lines: enzyme inhibition (c, competitive; a, allosteric). Stippled boxes: isomers whose concentrations were measured as pool. Abbreviations from Supplementary Table 1. (**c**) Typical parameter estimation experiment from the lower part of glycolysis (experiment E3 of Supplementary Table 12). Upper panel: summary of starting conditions and interventions during experiment. Units refer to the absolute amount of enzyme added at a given time, concentrations to expected concentration changes. Lower panel: blue, measured concentrations; green, simulation. (**d**) Affinity parameters with best estimate as blue star, estimation boundaries (green) and range of parameters mentioned in the literature (red).

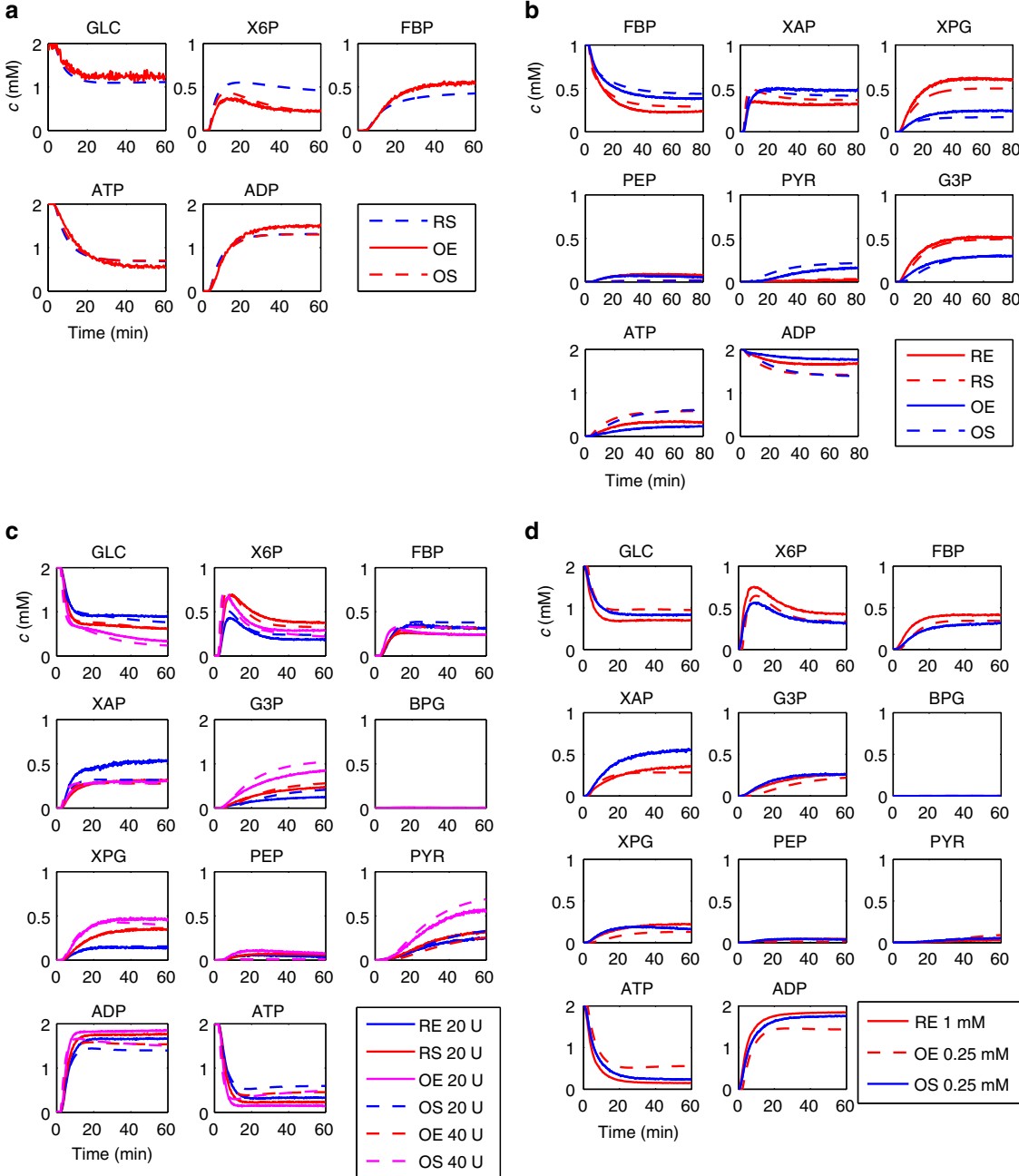

**Figure 3 | System optimization.** (**a**) Performance of equi-activity (reference, only predicted) and optimized (predicted and measured) upper part of reaction system. (**b**) Performance of equi-activity and optimized lower part of reaction system. (**c**) Performance of complete reaction system after distribution of either 20 or 40 U of enzymes, either in equi-activity or in optimized distribution. (**d**) Performance of complete reaction system (total of 10 U) with 0.25 or 1 mM NAD in the feed. All experiments were conducted with constant feed and started by injection of the respective enzyme system. E, experiment; O, optimization; R, reference; S, simulation.

implemented, G3P productivity is indeed hardly affected (Fig. 3d).

In summary, we demonstrate the non-iterative model-based forward design and implementation of one of the most complex *in vitro* enzymatic reaction systems ever implemented for chemical production. Clearly, forward design is possible even in highly feedback-controlled systems such as those built on glycolytic enzymes, if only sufficient data of sufficient information content can be provided, which we did in the presented set of experiments. While, strictly speaking, we show this only for the case of the continuous reaction, the obtained model can of course

also be used to design batch reaction. The applied set-up allows for full control and observation of experiments, and the presented methods are scalable and allow for the construction and measurement of even more complex reaction systems for applications in fine and bulk chemical processes. Some aspects of this work will also be useful for experiments in the *in vivo* domain. Even though we used here enzymes from different hosts and in purified form (eliminating potentially unknown interactions with additional effectors and competing substrates that would be available in a cell, as well as possible effects of protein complexes and intermediate and/or product sinks), the presented

model has been shown to capture the system behaviour rather accurately and might thus serve as a qualitative tool for supporting the analysis of intracellular dynamics of glycolysis.

## Methods

**Source of enzymes.** All enzymes except Glk were obtained as summarized in Supplementary Table 3. For reactor experiments the catalytic unit definition as provided by the supplier and as detailed in Supplementary Table 3 was used. Glk was produced as a His$_6$-tagged variant recombinantly in an *E. coli* BL21 strain from a gene under the control of the phage T7-promoter on pBR322-type plasmid and affinity purified. For more detailed description see Supplementary Methods.

**Set-up of experimental system.** Perturbation functions were generated by combining a (multi-channel) HPLC pump and an injection loop, connected through polyether ether ketone capillaries to the main reactor. Supplying defined compound concentrations in the reaction buffer from the HPLC pump established a continuous reaction feed; switching abruptly from one to a second reaction buffer allowed implementing step functions. Switching gradually over time allowed implementing a ramp function and even more complex input functions such as a sine could be realized by programming the pump. Using the injector for compounds allowed approximating an impulse by producing a pulse from an injector loop volume small in relation to the flux provided by the pump. Finally, using it for enzyme supply produced a step function in terms of reactor enzyme concentration. The enzyme membrane reactor (Bioengineering AG, Wald, Switzerland) was constantly stirred (600 r.p.m.) and featured a cellulose triacetate membrane with a cutoff at 20 kDa. The reactor effluent was conditioned for subsequent online MS detection by diluting it between 30- and 100-fold with the MS matrix buffer. Afterwards, the flow was split to limit influx into the MS. The exact fluxes were calculated from standard compounds (see below). For more detailed description see Supplementary Methods.

**Mass spectrometry.** A triple quadrupole mass spectrometer (MS) (MDS Sciex 4000 Q-TRAP, Applied Biosystems, CA, USA) with ESI was used in multiple reaction monitoring setting and in negative-ion mode. The applied routines (Supplementary Methods) did not allow resolving structural isomers, specifically the pairs G6P/F6P, 2PG/3PG and DHAP/GAP. These were measured in three pools: X6P (for G6P/F6P); XPG (for 2PG/3PG); and XAP (DHAP/GAP).

**Identification of experimental system.** The experimental system was decomposed into subunits, which were identified one after the other using 1 mM GLC in reaction buffer to generate perturbations recorded by the MS. Neglecting the set-up-specific dead times, the injector response was close to an ideal step. The other system elements generally behaved as first-order lag elements with dead time and were fitted to a CSTR model. For more detailed description including estimated volumes see Supplementary Methods.

**Compound quantification.** To measure accurate concentrations despite ion suppression effects and irregularities in flow through capillaries, we used three different procedures: the flux was monitored by adding different isotopologues of taurine to reaction and MS-matrix buffer, and ion suppression was accounted for by either adding isotopologues of starting materials, intermediates and products to the MS-matrix buffer to serve as a known reference measured under identical ESI conditions or through extensive calibration against HEPES as another standard in the MS-matrix buffer. For a detailed treatment of compound quantification see Supplementary Methods.

**Enzyme stability.** Enzyme stability was confirmed indirectly from the stability of steady-state signals for the various starting materials, intermediates and substrates in the CSTR during specific experiments. See Supplementary Methods for more details.

**General computational methods and kinetic model.** All computations were performed with Matlab (Mathworks, MA, USA). For efficient simulation the kinetic model was coded in C and compiled by the SBPD extension package of the Systems Biology Toolbox 2 (ref. 50) together with the CVODE integrator[51] as a Matlab-callable MEX function. This allowed for a, in comparison with Matlab, very fast model integration with up to 2.5e8 simulations of the kinetic model per day on a 12 core server system, assuming the least complex constant feed experiment with a single initial enzyme injection and 60 min simulation time. Thermodynamic data were calculated using eQuilibrator 1.0 (ref. 52). The kinetic model was constructed by balancing the compounds, formulating rate laws for the enzymes and embedding them into the identified three-vessel system of the experimental set-up. The kinetic model describes the reaction network on the basis of 54 ordinary differential equations derived by balancing the metabolite fluxes within the reactor. Rate equations were derived based on literature information considering known mechanism, sequence of metabolite binding and effectors. Reactions were modelled

as irreversible if the calculated thermodynamic equilibria were more than 1,000-fold on the product side. Special care was taken to include regulatory properties of the pathway members, such as allosteric effects, competitive inhibitions and cooperativity. For detailed information on equilibria and the model itself see Supplementary Methods.

**Optimization of fit.** To search the parameter space for parameter estimation and optimizing enzyme concentrations a Gaussian adaptation algorithm[53] served as optimizer. The used algorithm is a probabilistic method resulting in different outcomes for different optimization runs and so by default multiple runs were performed to ensure optimal results. For the estimation of parameter *p*, the mean squared error function between measurement and simulation was minimized as a quality criterion. For more details, see Supplementary Methods.

**Data processing.** The practical implementation of the compound quantification strategies required a standardized workflow to deal with the imperfection of real data, consisting of first, elimination of crosstalk in raw data; second, calculation of metabolite concentrations; third, correction of calculated concentrations by matching of observed and set concentrations; fourth, calculation of ideal and real mass balances; and finally corrections of calculated concentrations by matching with the ideal balances. The elaborate treatment of these steps is detailed in Supplementary Methods and Supplementary Fig. 13.

**Parameterization.** For the initial parameterization of the reaction model and the definition of parameter boundaries for the parameter estimation we used different sources. $V_{max}$ values were obtained from the known amount of added enzymes, equilibrium constants were calculated from thermodynamic data (eQuilibrator 1.0; ref. 52) as in Supplementary Table 23 and intrinsic enzymatic properties such as enzyme affinities ($K_M$ values) were obtained from databases such as BRENDA[49] and literature research. Then, improved estimates for the affinity parameters, thermodynamic equilibria and Hill coefficients were obtained by varying initial values and comparing simulated and experimental data and minimizing the difference. Where specific reactions could not be resolved (see pools above), affinities from literature were maintained. For detailed information on parameterization, see Supplementary Methods.

**System optimization.** The optimization of the reaction system was performed by adapting the activity of single enzymes for a given total amount of activity with regards to one- or two-compound concentrations. For this the optimizer suggested an activity distribution and simulated the experimental outcome of a constant feed experiment with initial enzyme injection for a chosen experimental length. The finally reached concentration at steady state was taken as criterion and provided to the optimizer that then suggested another distribution. For more detailed information see Supplementary Methods.

**Data availability.** Data supporting the findings of this study are available within the article (and its supplementary information files) and from the corresponding author on reasonable request. Also, the MATLAB model and all concentration data are available in the ETH Data Archive with the identifier IE2151530 (ref. 54).

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

## Acknowledgements

We thank Joerg Stelling for critical reading of the manuscript; Anne Femmer who provided assistance in many of the reactor experiments and the Glk production; and Hiroki Kawahara for the construction of the *glk* expression plasmid. We acknowledge financial support from The European Union (Projects EuroBioSyn (#12749), Nanomot (#29084) and ST-FLOW (#289326)), and the ESF-sponsored project Nanocell.

## Author contributions

C.H., S.B. and S.P. conceived the experiments; C.H. and S.B. performed the experiments; C.H. analysed the data; C.H. and S.P. co-wrote the paper.
