## [Peer Review File · Nature Communications]

Reviewer #1 (Remarks to the Author):

Summary:

This article describes the forward engineering of a complex biosynthetic pathway comprising ten enzymatic steps. The manuscript is well written, the data are impressive, and the modeling supports their data. The combination of these elements will make this a potentially impactful paper. The authors used purified enzymes to implement a multi-step enzymatic cascade, along with continuous substrate flow in a CSTR setting. Metabolic concentrations are measured in a detailed way and these data are used to build models and predict pathway operation. This technology has applications in providing insights into the operation of enzymatic pathways, including bottleneck points. I believe the implementation for forward engineering at this scale is new. While this approach will teach us about metabolic pathways, the continuous operation and flow of cofactors would make it economically infeasible for synthesis applications. Along these lines, my major concern is that the authors select pathways for DHAP and G3P synthesis. DHAP is the same molecule used in the authors' previous 2011 Nat. Chem. Bio. paper, and G3P is one step further. This pathway is well studied, and selecting a less understood pathway would have increased impact and novelty. Is there a possibility that this approach can be used to refine pathways in cells? Can we learn about bottlenecks using this system and then transfer that knowledge to a more complex environment? Despite these concerns lessening my enthusiasm, I believe this is an important paper. Some additional concerns to be addressed are below.

Concerns:

Can the authors please clarify the x-axis in Figure 2C? The pulses occur at non-standard intervals and it would be useful to indicate or reflect the system alterations in the traces of data somehow. As it stands, it is difficult to work through the data and pathway response.

The parameter estimation in Figure 2D is confusing and needs clarification. The authors state "we ensured accurate measurement of compound concentrations despite the potential for ion suppression due to the concomitant entry of many compounds into the ESI chamber of the MS." Then the authors state, "The remaining 8 parameters showed values close to the boundaries we had set for parameter estimation, indicating that more optimal values might have been obtained with better data." I am confused why the MS isn't providing good data, if the authors ensured accurate measurements. Can the authors clarify and expand upon the limitations? The authors note that they just stop trying to improve the system as their model was already good enough for forward engineering. How do they know that beforehand? It is not clear to the reader that this model is good with the what appears to be messy parameter estimation.

Given the detailed nature of the data, can the authors clarify what benefits the model provides? In other words, could similar predictions simply have been made by "looking" at the data?

The authors mention economic improvements by reducing the NAD concentration 4-fold. What is the impact of this in a more comprehensive economic analysis?

Is the forward design only possible in this purified enzyme cascade system? Can the authors please comment on extension to more complex systems, such as a cell, where general metabolism and network interconnectivity will be affecting each of these steps in different ways? If the approach does not translate to cells, what types of cascade systems would it make economical sense to pursue using this approach, especially given the continuous nature and waste of cofactors? How would one translate the information from the continuous flow system to a batch system for example, where expensive cofactors would not be flushed out of the system?

Reviewer #2 (Remarks to the Author):

This research paper first reports on a novel experimental setup that allows perturbation and tracking of systems dynamic behaviour of a complex synthetic enzyme cascade reaction (with 10 enzymes and 18 metabolites). The setup is equipped with an on-line mass spectrometry and a continuous operation system for applying system theory input functions to dynamically perturb the system. The high data-density obtained under dynamic conditions enables a thorough model identification and parameterization. For the first time a 10-enzyme reaction cascade has been forward designed and optimized in a model-guided manner. The work is original. It represents a significant progress towards design of synthetic enzyme cascades which are of great interests not only for the development of cell-free biosynthesis systems, but also for implementing new or improving existing metabolic pathways *in vivo*.

The novel approach is clearly described. A vast amount of high quality data was generated and used for setting up the model and for system optimization. The presentation is in general fine. But the size of the graphs is quite small and the content is so dense in each of the Figures in the main manuscript, that it is not easy to follow. Specifically, the authors may consider the following comments/suggestions for improving the quality of presentation.

1. Fig.1C can be hardly understood without referring to the manuscript and Supplementary Materials for many times.

2. The legend for Fig.2 A should be also made very clearly.
3. In Fig.2D several parameters are far outside of the boundary. In the text (bottom of page 4) the authors stated that "the remaining 8 parameters showed values close to the boundaries...". PI check it again and rephrase the sentence.
4. The authors should also check the legend for Fig.3A (colors of the lines in the box).
5. Concerning the results presented in Fig.3 why the authors didn't carry out the experiment in D also with 20 or 40 U enzyme for a better comparison?

The points mentioned above do not affect the conclusions of the authors. They are robust and valid.

References are adequate. The Abstract and the text are well written. In conclusion it is an original and innovative work with large impacts on the field of cell-free biosynthesis and synthetic biological systems. It is well suitable for publication in Natural Communications.

Reviewer #3 (Remarks to the Author):

The paper presents the experimental in vitro implementation of a cell-free system that includes (almost) the complete glycolytic pathway.

The authors have done a great job where (i) they implemented a complex reaction system, (ii) they developed a mathematical model that describes the enzyme kinetics and the dynamics of the metabolites, (iii) they identified the parameters of the model, and (iv) they used the mathematical model to optimise the reaction network and identify approaches for optimal enzyme and cofactor utilisation for the operation of the system.

The work is technically very good and the overall approach can be used for other cell free systems. However the approach and the learnings here cannot be used for optimisation of pathways in vivo as the authors claim. The pathways in vivo operate under much more complex conditions and the in vivo analysis can be misleading instead of helpful. The simple and important case of NADH utilisation by many more enzymes in vivo suggests that the in vivo system studied here is an oversimplification.

The authors also list a number of advantages of using a cell free system and they discuss about the use of such systems for the production of a number

of biochemicals (even bulk chemicals!). The cell free systems remain an unproven system for industrial production and definitely not a competitive technology and feasible for large-scale production of commodity (bulk) chemicals.

There is also significant and important earlier work in similar systems that the authors either are not aware of or choose not to cite.

For example the excellent work by Adam Arkin and John Ross (e.g., *Science*, 1997).

Response to referees

Manuscript "Forward design of a complex enzyme cascade reaction" by Hold et al.

General comment: As will become clear from the comments below, most of the critical points referred either to a limited novelty in the particular chemical system that we selected (1), a limited interest for in vitro systems in general (2), or insufficient transferability to in vivo systems (3). We would like to respond to these more general comments upfront in a centralized way, before we address the comments point-by-point. Changed text is marked in yellow, reviewers in blue.

(1) While we agree that the chemical nature of the reaction system is an important point, we argue that focusing on this element misses what we consider to be the essential element of the manuscript, which is a demonstrated example of biological design at an unprecedented scale. We present is a framework of experimental analysis, dynamic excitation and response measurement of cell free enzymatic systems allowing the parameterization of dynamic models with predictive usage.

In order to achieve this, we had to implement the experimental and theoretical foundations of the required engineering analysis, and here – in the demonstration that, with a suitable analysis of the system at hand in a novel experimental framework coupled to rigorous use of modeling techniques, design of even complex biosystems becomes possible – we think the novelty lies.

The implementation of this experimental frame was challenging, and we argue this more than justifies the selection of a reasonably well understood system for a first implementation. After all, there is no documented case of forward design using this system (glycolysis in a wider sense) that we are aware of – and that despite the fact that glycolysis has been a model system since the 1960's. Therefore, even though we undoubtedly know much about the system, we argue that the understanding of the system remains severely limited – in fact so much that design has been impossible until now.

*(2) In order to illustrate that the approach taken here is not an esoteric exercise but advances a field of potential and real biotechnological significance, we argued the point in the manuscript that in vitro applications are gaining considerable traction. While it is of course true that we took the liberty of selecting a broad range of examples for the applications of cell free cascade reactions, there are still a lot of examples of practical import to consider. In fact, we feel a compelling case for the potential economic import of cascade reactions has been made in two recent summary papers from the group of Perceival Zhang: "New-biotechnology paradigm: cell-free biosystems for biomanufacturing", Rollin, Tam and Zhang, *Green Chemistry* 15:1708-1719 (2013), and "Biofuel production by in vitro synthetic enzymatic pathway biotransformation", Zhang, Sun and Zhong, *Current Opinion in Biotechnology* 21:663-669 (2010). As convincingly laid out there, cell-free systems can be used efficiently even for bulk products (eg ethanol). We do not have much to add to this except for pointing out that we feel that the case becomes even stronger when looking at the fine chemical industry (in which the senior author had the privilege to work during the earlier stages of his career): In particular in the field of oligosaccharides and unnatural monosaccharides, but also in the manufacturing of enantiomerically or diastereomerically pure compounds, cell free cascade reactions have been essential or are becoming important (see the references also in the present manuscript and below). Given the time pressure, for example in the production of pharmaceutical intermediates, methods that allow rapidly taking a complex reaction system to optimal operation (instead of conducting lengthy combinatorial*

experimental campaigns) are of high value. In this manuscript, we provide such a clear path to implement such an optimized operation with a much reduced experimental effort.

(3) Regarding the transfer to *in vivo* systems: To state it explicitly, the present manuscript presents an approach for the design of an *in vitro* pathway, and we feel that the conceptual advance (experimental framework, algorithms, demonstration of design) is substantial and, in our view, the core of what should be evaluated in this manuscript. However, we cannot deny that it is tempting to make the connection to metabolic engineering/*in vivo* pathway engineering. We were therefore careful in the submitted version of the manuscript not to suggest that the method as presented here can be used in the same way for *in vivo* applications. However, in order to further remove any potentially misleading comments, we have further clarified this by changing the manuscript in the following way

- a) We changed the first sentence of the last paragraph of the introduction to read “We reasoned that **in vitro** forward design would become possible if the experimental system allowed a sufficiently broad application of dynamic challenges and a sufficiently detailed recording of the system’s responses.”
- b) We removed the last sentence of the conclusions.

Having said this, we of course maintain that the model as such will be useful for any kind of qualitative consideration of *in vivo* glycolysis, as such models have been useful over the last decades in the interpretation of unexpected behavior of cells, and as such might support future *in vivo* endeavors. However, the model will only be a guide to explore possibilities, for the following reasons:

- a) The enzymes we used were from different sources. To increase usefulness, all enzymes would need to come from one source, the organism of interest.
- b) Even if that were the case, the model would not be an accurate reflection of what is going on in the cell, because of:
 - additional unknown effectors *in vivo*,
 - additional competing substrates *in vivo* (eg. GTP and other nucleoside phosphates),
 - the presence of protein complexes *in vivo* that would not have been reflected in our setup,
 - compound drains by competing reactions/pathways *in vivo*
 - enzyme concentration level changes in response to regulation *in vivo*

Finally, there is the question whether the approach as such could be transferred to an *in vivo* situation. This would be very difficult, as of course—taking *E. coli* as an example—of the investigated compounds only glucose can cross the membrane, and this only with a more reduced dynamic than used in the experiments here, which prevents proper parameterization of the model. Therefore, it seems more realistic to install gene circuits that enable specific perturbations (such as oscillations) in the cell, although this would limit the approach on the analysis side to cells that can be “synchronized” (eg yeasts). The problem of measuring *in vivo* metabolite concentrations in high frequency could most probably be overcome.

Despite these links, we would like to refrain from commenting on them in the manuscript, as the main point of the paper is not preparing the ground for a later *in vivo* application, but to experimentally demonstrate that forward design is possible in a system of this complexity, provided a suitable experimental framework is established. We are aware of the temptation to link this work to *in vivo* questions (see the comments of the reviewers below), but we feel that a comprehensive discussion of this aspect would shift the focus too much away from our fundamental point.

In summary, we do not argue that this approach is transferable, because there are quite a number of issues, as discussed above. However, we do maintain that the model as such is very valuable in giving qualitative guidance for in vivo questions relating to glycolysis.

Reviewer #1 (Remarks to the Author):

Summary:

This article describes the forward engineering of a complex biosynthetic pathway comprising ten enzymatic steps. The manuscript is well written, the data are impressive, and the modeling supports their data. The combination of these elements will make this a potentially impactful paper. The authors used purified enzymes to implement a multi-step enzymatic cascade, along with continuous substrate flow in a CSTR setting. Metabolic concentrations are measured in a detailed way and these data are used to build models and predict pathway operation. This technology has applications in providing insights into the operation of enzymatic pathways, including bottleneck points. I believe the implementation for forward engineering at this scale is new.

We thank the reviewer very much for her of his kind words.

While this approach will teach us about metabolic pathways, the continuous operation and flow of cofactors would make it economically infeasible for synthesis applications.

Of course, the reviewer is right and upon reading the manuscript once more, we realized that we did not sufficiently clarify the role of the CSTR as an important step in the engineering analysis rather than as a production regime. We therefore inserted a clarifying sentence in the results section: "While a CSTR is not an optimal way to operate such a multistep reaction, it is indispensable for a thorough engineering analysis of the system."

Having said this, we would like to point out that the mode of analysis requires a CSTR. We are continuously measuring system composition, which requires a bleed stream from the reactor. As this measurement should not influence the volume (which is of course an important system parameter), then either the reactor has to be very large (making the experiments very expensive) or it has to be operated as a CSTR, which is the option that we selected.

Along these lines, my major concern is that the authors select pathways for DHAP and G3P synthesis. DHAP is the same molecule used in the authors' previous 2011 Nat. Chem. Bio. paper, and G3P is one step further. This pathway is well studied, and selecting a less understood pathway would have increased impact and novelty.

The reviewer addresses an important point, the selection of the model system. While it is correct to point out that this is similar to the system used in our earlier report, we would like to return to the point we tried to make in the general comments above: We argue that the crucial achievement of the reported work is biological design on an unprecedented scale, which required novel developments in experimental framework and theoretical analysis. Therefore, we felt that we were justified in selecting a system about which much is known – though, tellingly, not enough to show one single case of successful forward design that we are aware of.

Is there a possibility that this approach can be used to refine pathways in cells? Can we learn about bottlenecks using this system and then transfer that knowledge to a more complex environment?

We addressed this point also in the general comments. Here, we would like to point out that in vitro systems are an important field of application in itself. Therefore, our main point is design of an in vitro cascade reaction, and we feel that this goal is well justified. Regarding the transfer to in vivo situations, we refer to the general comments.

Despite these concerns lessening my enthusiasm, I believe this is an important paper.

Again, we would like to thank the reviewer for her/his kind words.

Some additional concerns to be addressed are below.

Concerns:

Can the authors please clarify the x-axis in Figure 2C? The pulses occur at non-standard intervals and it would be useful to indicate or reflect the system alterations in the traces of data somehow. As it stands, it is difficult to work through the data and pathway response.

We have clarified the axis in 2c, which was time in all examples according to the model graph in the lower left corner.

The parameter estimation in Figure 2D is confusing and needs clarification. The authors state "we ensured accurate measurement of compound concentrations despite the potential for ion suppression due to the concomitant entry of many compounds into the ESI chamber of the MS." Then the authors state, "The remaining 8 parameters showed values close to the boundaries we had set for parameter estimation, indicating that more optimal values might have been obtained with better data." I am confused why the MS isn't providing good data, if the authors ensured accurate measurements. Can the authors clarify and expand upon the limitations? The authors note that they just stop trying to improve the system as their model was already good enough for forward engineering. How do they know that beforehand? It is not clear to the reader that this model is good with the what appears to be messy parameter estimation.

We are sorry for the misleading wording. We insist that the data quality from MS and the quantification system is quite accurate. However, while we showed that the experimental setup is useful in resolving difficult cases of structural identifiability (see glucokinase case), we did not resolve all such difficult instances in the model, as it simply was not required for the design. Consequently, the parameters from these non-resolved cases are likely to remain at the boundaries selected for estimation. Some of the confusion might stem from the fact that in addition to the boundaries for de novo estimation of parameters, we also discuss parameters ranges that stem from a literature survey. We integrated both sets of boundaries into Fig. 2D and adapted the manuscript text as follows:

*"Ultimately, the set of 22 experiments allowed **estimating** a final set of parameters (Table S19 **and Fig. 2D**) that reproduced all experiments well as the basis for the subsequent design phase. **Only few parameters (e.g. for Pfk) were at the boundary of the allowed range, suggesting cases of limited identifiability that might have been resolved with additional experiments.** Next, consultation of suitable databases (in particular BRENDA⁴⁸) allowed comparing 32 of the 36 compound affinities,*

whose knowledge we had deemed most uncertain before, with previously known values. Of these 32 affinities, 24 were within generally reported ranges (Fig. 2D). Possible reasons include that our reaction conditions might have been different from those under which these parameters had been previously established (in particular, presence of a broad variety of compounds). However, we refrained from further refining the perturbations and the parameter set as the models for the subsystems as well as the complete model were already fully capable of enabling forward design."

Given the detailed nature of the data, can the authors clarify what benefits the model provides? In other words, could similar predictions simply have been made by "looking" at the data?

Modeling is advantageous when either the system is too complex for intuitive optimization or when quantitative predictions are required. Even though we have been working with the system for several years, we do not feel capable of intuitively optimizing this 10-enzyme system (which is of course why we undertook to construct the experimental framework that is described in this manuscript – let it suffice to mention here the interlaced regulation scheme, e.g. when FBP is a Pfk product that allosterically activates Pyk, whose substrate PEP allosterically inhibits Pfk). We definitely cannot intuitively predict the precise distribution of enzymes that would lead to optimal behavior. This can only be predicted quantitatively with a model.

The authors mention economic improvements by reducing the NAD concentration 4-fold. What is the impact of this in a more comprehensive economic analysis?

NAD is the single most important cost entry on the chemical side of the system (on a cost per g or kg basis: the costs for glucose and phosphate are negligible, ATP can be obtained for 300 \$/kg (Kyowa Hakko), NAD for 1700 \$/kg. Of course, in the setup selected here, the overall most important cost entry is enzymes. However, if this system was implemented on large scale, the enzymes would not be commercially acquired as we did for the work here, but the system would be reengineered and most likely switched to thermophilic (see the cost analysis in Current Opinion in Biotechnology 21:663-669) or immobilized enzymes, at which point the cost contribution of the enzymes would drastically decrease. Therefore, we insist that NAD supply remains in the long-term the single most important cost entry of the system.

Is the forward design only possible in this purified enzyme cascade system? Can the authors please comment on extension to more complex systems, such as a cell, where general metabolism and network interconnectivity will be affecting each of these steps in different ways?

In general, the forward design is possible for any cascade that operates under a similar set of conditions, also considerably larger cascades (if analysis is parallelized). A precondition is that it has to be possible to calibrate the MS-analysis, and for some pathways not all the intermediates might be available without additional efforts (eg self-production of intermediates, etc).

For the extension to cells we refer to the general comments.

If the approach does not translate to cells, what types of cascade systems would it make economical sense to pursue using this approach, especially given the continuous nature and waste of cofactors? How would one translate the information from the continuous flow system to a batch system for

example, where expensive cofactors would not be flushed out of the system?

As pointed out above, the reaction system is not expected to be applied in a CSTR, but in a batch reactor. We clarified this also in the text (see above).

In general, there are two areas for applications, bulk and fine chemistry. One obvious application is ethanol production, where in vitro or quasi in vitro approaches using thermophilic enzymes are increasingly used (see Current Opinion in Biotechnology 21:663-669). We already pointed out the overview work of Zhang, in which the economic case is (in our view) convincingly made.

The second, maybe not so obvious, area is fine chemistry, i.e. the synthesis of multifunctional intermediates, mostly for pharmaceutical purposes, but also for applications such as in the flavor and fragrances industry. Cascade reactions become increasingly important here, as pointed out in the various references in the present manuscript (eg Sehl 2013, O'Reilly 2014), but also in review articles such as Guterl 2012, Guterl 2013, Billerbeck 2013, Dudley 2015, all cited in the manuscript). We would also like to point out that in particular the synthesis of oligosaccharides and unnatural monosaccharides is essentially a history lesson in cascade reactions, including commercial applications (e.g. the synthesis of activated monosaccharides by Kyowa Hakko, Endo 2001 in this manuscript, or Fessner 2015 for a recent review).

As for the transition from continuous to batch operation, the answer is straightforward – the transition would be guided by the model. The most favorable scenario for a specific batch can be simulated by only changing one parameter in the current model (the dilution rate is simply set to zero). We also simulated this case, but due to the requirement for a constant effluent we cannot easily realize this situation in our setup and so the simulation would remain experimentally unconfirmed. Intermediate scenarios such as reducing effluent flow or increasing reactor volume can be imagined, but would not change the situation fundamentally. Furthermore, we should point out that, given our reactor volume of about 7 ml and a flux of 0.25 ml/min, the mean residence time is about 28 minutes which is in contrast to the much faster reaction dynamics, and thus effectively our conditions are relatively close to batch conditions.

However, the main point regarding this comment is that the model can be used to easily identify the optimal batch scenario.

Reviewer #2 (Remarks to the Author):

This research paper first reports on a novel experimental setup that allows perturbation and tracking of systems dynamic behaviour of a complex synthetic enzyme cascade reaction (with 10 enzymes and 18 metabolites). The setup is equipped with an on-line mass spectrometry and a continuous operation system for applying system theory input functions to dynamically perturb the system. The high data-density obtained under dynamic conditions enables a thorough model identification and parameterization. For the first time a 10-enzyme reaction cascade has been forward designed and optimized in a model-guided manner. The work is original. It represents a significant progress towards design of synthetic enzyme cascades which are of great interests not only for the development of cell-free biosynthesis systems, but also for implementing new or improving existing metabolic pathways in vivo.

The novel approach is clearly described. A vast amount of high quality data was generated and used for setting up the model and for system optimization. The presentation is in general fine.

We would like to thank the reviewer for her or his kind words.

But the size of the graphs is quite small and the content is so dense in each of the Figures in the main manuscript, that it is not easy to follow. Specifically, the authors may consider the following comments/suggestions for improving the quality of presentation.

We thank the review for the comment. The limited quality is mainly to due to the integration of the figures in Word For print purposes high resolution graphics will be provided allowing to view the data in detail.

In addition to the specific requests addressed below, we also relaxed Fig. 2C. In the previous versions, two experiments were shown, one for parameter estimation of the upper pathway, one for the lower. As both experiments are very similar in type and message, we decided to remove one experiment from the figure to improve readability. The removed experiment continues to be part of the supplementary material.

1. Fig.1C can be hardly understood without referring to the manuscript and Supplementary Materials for many times.

We changed the details and the legend of Figure 1C to address this issue and hope that it is sufficiently clear now.

2. The legend for Fig.2 A should be also made very clearly.

We changed the legend of Figure 2A to address this issue and hope that it is sufficiently clear now.

3. In Fig.2D several parameters are far outside of the boundary. In the text (bottom of page 4) the authors stated that "the remaining 8 parameters showed values close to the boundaries...". Pl check it again and rephrase the sentence.

We apologize for the confusion and point to the response to the comments of reviewer 1, where we treat this point comprehensively.

4. The authors should also check the legend for Fig.3A (colors of the lines in the box).

Done

5. Concerning the results presented in Fig.3 why the authors didn't carry out the experiment in D also with 20 or 40 U enzyme for a better comparison?

There is no specific reason. We think we have demonstrated that the model is quite reliable across a broad range of total enzyme amounts in the system (see Fig. 3c), and the optimization potential was in our view clearly demonstrated already for the 10 U case. Therefore, we decided to leave it at that.

The points mentioned above do not affect the conclusions of the authors. They are robust and valid. References are adequate. The Abstract and the text are well written. In conclusion it is an original and innovative work with large impacts on the field of cell-free biosynthesis and synthetic biological

systems. It is well suitable for publication in Natural Communications.

Once again, we would like to thank the reviewer for her or his kind words.

Reviewer #3 (Remarks to the Author):

The paper presents the experimental in vitro implementation of a cell-free system that includes (almost) the complete glycolytic pathway.

The authors have done a great job where (i) they implemented a complex reaction system, (ii) they developed a mathematical model that describes the enzyme kinetics and the dynamics of the metabolites, (iii) they identified the parameters of the model, and (iv) they used the mathematical model to optimise the reaction network and identify approaches for optimal enzyme and cofactor utilisation for the operation of the system.

The work is technically very good and the overall approach can be used for other cell free systems.

We would like to thank the reviewer for his or her kind words.

However the approach and the learnings here cannot be used for optimisation of pathways in vivo as the authors claim. The pathways in vivo operate under much more complex conditions and the in vivo analysis can be misleading instead of helpful. The simple and important case of NADH utilisation by many more enzymes in vivo suggests that the in vivo system studied here is an oversimplification.

We are not quite sure how to address this issue. We carefully re-read the manuscript several times in order to find out where in the text we claim that our method can be used for optimization of in vivo systems. The only section of which we are aware in which we address this issue is the last sentence of the conclusions, which reads "By ensuring that all enzymes come from only one host, additional insight into system properties might also support the optimization of pathways in vivo"⁴⁹. " The point of this sentence is to acknowledge that we collected enzymes that were isolated from different sources and that therefore the model cannot be used as a qualitative representation of eg E. coli or S. cerevisiae metabolism. Had we had access to all enzymes from one source, then the model might of course have been helpful in answering qualitative question about possible changes of eg E. coli glycolysis in view of a specific application. We fully agree that the extent of the usefulness might be limited by the fact that we parameterized it under in vitro conditions etc (see also the more detailed discussions above, in the general section and regarding reviewer 1). Still, also qualitative support is "support". However, the main argument is that the present manuscript does not make the point that we can do in vivo pathway optimization this way, it merely points out that under specific circumstances in vitro models obtained in this manner can "support" in vivo optimization. To eliminate possible sources of misunderstanding, we changed the first sentence of the last paragraph of the introduction to read "We reasoned that in vitro forward design would become possible if the experimental system allowed a sufficiently broad application of dynamic challenges and a sufficiently detailed recording of the system's responses." Furthermore, we removed the last sentence of the conclusions.

The authors also list a number of advantages of using a cell free system and they discuss about the

use of such systems for the production of a number of biochemicals (even bulk chemicals!). The cell free systems remain an unproven system for industrial production and definitely not a competitive technology and feasible for large-scale production of commodity (bulk) chemicals.

We respectfully disagree with the reviewer. Cell free systems are of course used for bulk chemical productions (high fructose corn syrup, acrylamide, etc). Cascade reactions not yet, but the economic case has been made (see model calculations by Zhang 2010) and a variety of attempts have been made to address alcohol production cell free with thermophilic enzymes (eg Krutsakorn 2013, Guterl 2012, cited in the manuscript). Some of these attempts include very serious chemical companies (Südzucker, now Clariant). Even if that were not the case, we argue that there is a long history of using cell free cascades for preparative purposes in chemistry, in particular in the field of oligosaccharides and unnatural monosaccharides, as illustrated – among a broad range of other examples – by the success of the commercial Kyowa Hakko technology for providing activated monosaccharides.

There is also significant and important earlier work in similar systems that the authors either are not aware of or choose not to cite.

For example the excellent work by Adam Arkin and John Ross (e.g., Science, 1997).

This is indeed embarrassing. Despite the fact that we have worked in the field of cascade reactions for 12 years and work on the design aspect for 6, we have to admit that we were not aware of this paper. We are therefore deeply grateful to the reviewer for pointing out this work to us, and of course we integrated this important reference in the revised text. Specifically, we cite the work when discussing the setup at the beginning of the Results and Discussion section.

Having said this, we would like to point out that the purpose of the work by Arkin et al was (black box) model identification of unknown interactions, not design. Based on such information possible interactions can be formulated and decided upon. In the present work, we aimed at the other end of model purposes and worked on the next step, namely parameter identification for a system of known interactions by valuing their strength, for design purposes. We acknowledge that the work by Arkin et al is indeed applying a CSTR for their analyses as well, but we argue that the experimental framework presented in the present manuscript is significantly further developed (in view of structure of the perturbation and integrated analytics) to justify publication in its own right.

Response to referees

Manuscript "Forward design of a complex enzyme cascade reaction" by Hold et al. (NCOMMS-16-03683B)

Responses to issues raised by referees

REVIEWERS' COMMENTS:

Reviewer #1 (Remarks to the Author):

The revised manuscript by Panke and colleagues titled "Forward design of a complex enzyme cascade reaction" is an important demonstration of how in vitro enzyme systems can be used in biological design. The authors emphasize in their rebuttal that this should be our key take away message. To this point, I agree the work is a significant achievement and novel tour de force. I do not believe the authors need to demonstrate transfer to in vivo systems. The advance reported has broad interest in industrial and academic research and should make possible new applications in metabolic engineering and synthetic biology. I also align with the authors with the idea that in vitro cascade reactions could have a significant economical impact for fine chemicals.

In the revised manuscript, the authors addressed my previous concerns and the work has been improved. Although I would have liked a different model system, I do agree that the combination of experiments and theoretical work performed is impressive and unique. Put another way, the theoretical work here is new, thorough, and well-done. The design at this scale is also impressive, but I want to know that this would work for a new pathway system - perhaps that is the next paper?

Though many comments were addressed in the rebuttal, several of these were not integrated into the paper. I believe it is important to transfer many of the comments from the rebuttal to the manuscript.

We thank the reviewer for her or his kind comments.

Here are a few examples:

- The authors general comment #3 should be in some form included in the main text of the manuscript. If published, a majority of the readers will want to know the work's relation/applicability to in vivo pathway engineering. Being upfront about the promise and limitations of the work would be more helpful than being completely removed as was done in the revision.

Regarding general comment 3, which discusses possible links of this work to in vivo work, we added the following modifications:

We introduced the words “in vitro” into the abstract in order to make the focus on in vitro clear from the very beginning.

Furthermore, we transferred the following part from our general comment in the response letter into the last results paragraph. Here, we point out the limitations of our approach for in vivo work:

“Even though we used here enzymes from different hosts and in purified form (eliminating potentially unknown interactions with additional effectors and competing substrates that would be available in a cell as well as possible effects of protein complexes and intermediate and/or product sinks), the presented model has been shown to capture the system behavior rather accurately and might thus serve as a qualitative tool for supporting the analysis of intracellular dynamics of glycolysis.”

- In the text the authors state in the section about forward engineering to reduce NAD concentrations that their motivation is "Next to productivity, cofactor costs are a major impediment to the implementations of cascade reactions." Yet, the entire CSTR setup would also be a major impediment. The reviewers address this in the response with, "As pointed out above, the reaction system is not expected to be applied in a CSTR, but in a batch reactor." Yet there is no comment in the text explaining the comparison of CSTR in batch. The authors continue in the response with, "As for the transition from continuous to batch operation, the answer is straightforward - the transition would be guided by the model. The most favorable scenario for a specific batch can be simulated by only changing one parameter in the current model (the dilution rate is simply set to zero). We also simulated this case, but due to the requirement for a constant effluent we cannot easily realize this situation in our setup and so the simulation would remain experimentally unconfirmed." This whole section in the response should be discussed in the text as it pertains to the applicability and the usefulness of the analysis presented to the in vitro community.

Regarding the comments whether a CSTR is a suitable setup for carrying out a large scale reaction, we modified the manuscript as follows:

We integrated the following sentence into the first paragraph of the results section to clarify the use of the CSTR as a tool:

“While a CSTR would not be a suitable reactor for large scale implementation of a multi-step reaction, we reasoned that the setup would in fact remove the central obstacle for forward-engineering of complex reaction systems, namely limited model scope due to insufficient experimental data quantity and quality, and thus enable generating a model for design.”

Then, in the first section where we deal with design, we elaborate as follows: “Please note that, fundamentally, such a cascade would ideally be run in a batch or fed-batch reactor in order to prevent loss of intermediates (and thus a reduction in yield on glucose). However, in order to maintain the ability to track the behavior of the optimized system accurately and at high time resolution, we remained in the CSTR setting, which is anyway close to a batch scenario as the dilution rate is rather low (2.1 h^{-1}).”

Finally, we comment in the final paragraph (the discussion paragraph): "While, strictly speaking, we show this only for the case of the continuous reaction, the obtained model can of course also be used to design batch reaction."

- If the newly designed conditions for NAD do not follow the same trends in a batch reaction, then the forward engineering of these systems in this manuscript would not be as helpful for implementing in vitro cascade reactions. The added comment on why CSTR setup was used was great, but another comment needs to be in the text how the engineering analysis done in CSTR relates to batch reaction, the typical setup for in vitro systems in the field.

We feel that this part is already addressed in the modifications for the previous points.

Overall, the manuscript is well-done and technically sound. I remain convinced this is an important paper.

The reviewer quoted these points as "examples", suggesting that there might be more points we should transfer from our response letter to the discussion. We therefore went carefully through our responses to reviewer 1 once more and we argue that we covered the most important points. Specifically, "General comment 1" is an argument about the scope of the paper from our view (design), and we argue that this is exactly what we write in we tried to make this clear in the present manuscript. Similarly, "General comment 2" is a support for the importance of in vitro systems, which again is broadly discussed in the introduction. The remainder of our responses to reviewer 1 was to answer technical problems. Therefore we argue that we covered the salient points that reviewer 1 wanted to see in the main text.

Reviewer #2 (Remarks to the Author):

The authors have addressed all my questions convincingly in their rebuttal and done revision properly. This is an original work with novelty. It can be accepted for publication.

We thank reviewer 2 for her or his kind words.